# Diversity and Anti-Infectious Components of Cultivable Rhizosphere Fungi Derived from Three Species of *Astragalus* Plants in Northwestern Yunnan, China

**DOI:** 10.3390/jof10110736

**Published:** 2024-10-24

**Authors:** Guo-Jun Zhou, Wei-Jia Xiong, Wei Xu, Zheng-Rong Dou, Bo-Chao Liu, Xue-Li Li, Hao Du, Hai-Feng Li, Yong-Zeng Zhang, Bei Jiang, Kai-Ling Wang

**Affiliations:** 1Yunnan Key Laboratory of Screening and Research on Anti-Pathogenic Plant Resources from Western Yunnan, Institute of Materia Medica, College of Pharmacy, Dali University, Dali 671000, China; yaoxuewsw@163.com (G.-J.Z.); 13577544542@163.com (W.-J.X.); 17716868566@163.com (W.X.); dabolchao@163.com (B.-C.L.); 19187041748@163.com (X.-L.L.); aguyanyi111@163.com (H.D.); lihfzh888@sina.com (H.-F.L.); cell_zyz@163.com (Y.-Z.Z.); dalinorthjiang@163.com (B.J.); 2Key Laboratory of Feed Biotechnology, Ministry of Agriculture and Rural Affairs, Institute of Feed Research, Chinese Academy of Agricultural Sciences, Beijing 100081, China; 17864883192@163.com

**Keywords:** *Astragalus*, rhizosphere fungi, species diversity, secondary metabolite, antibacterial activity, antimalarial activity, special eco-environment

## Abstract

*Astragalus*, a group of legume plants, has a pronounced rhizosphere effect. Many species of *Astragalus* with limited resource reserves are distributed in the high-altitude area of northern Yunnan, China. Although some of these plants have high medicinal value, the recognition of them is still at a low level. The aim of this research is to explore the species diversity of cultivable rhizofungi derived from *Astragalus acaulis*, *A. forrestii* and *A. ernestii* growing in a special high–cold environment of northwest Yunnan and discover anti-infective components from these fungi. A total of 93 fungal strains belonging to 38 species in 18 genera were isolated and identified. Antibacterial and antimalarial screening yielded 10 target strains. Among them, the ethyl acetate crude extract of the fermented substrate of the rhizofungus *Aspergillus calidoustus* AA12 derived from the plant *A. acaulis* showed broad-spectrum antibacterial activity and the best antimalarial activity. Further chemical investigation led to the first discovery of seven compounds from the species *A. calidoustus*, including sesterterpine 6-epi-ophiobolin G; three sesquiterpenes, penicisochroman A, pergillin and 7-methyl-2-(1-methylethylethlidene)-furo [3,2-H]isoquinoline-3-one; and three polyketides, trypacidin, 1,2-seco-trypacidin and questin. Among them, the compound 6-epi-ophiobolin G exhibited moderate to strong antibacterial activity against six Gram-positive pathogens with the minimum inhibitory concentration (MIC) ranging from 25 to 6.25 μg/mL and a prominent inhibitory effect on the biofilm of *Streptococcus agalactiae* at an MIC value of 3.125 μg/mL. This compound also displayed potent antimalarial activity against *Plasmodium falciparum* strains 3D7 and chloroquine-resistant Dd2 at the half-maximal inhibitory concentration (IC_50_) values of 3.319 and 4.340 µmol/L at 72 h, respectively. This study contributed to our understanding of the cultivable rhizofungi from characteristic *Astragalus* plants in special high–cold environments and further increased the library of fungi available for natural anti-infectious product screening.

## 1. Introduction

The rhizosphere refers to a narrow region of soil and is usually considered a special eco-environment containing up to 10^11^ microbial cells [1,2]. These rhizosphere microbial communities are diverse and are of great significance to the process of growth regulation, defense and stress tolerance, as well as the quality of medicinal plants [3]. The application of metagenomic sequencing technology enables researchers to gain a macro understanding of the species composition and functional genes of microbes in special environments [4]. However, it is necessary to obtain pure cultured strains using traditional culture-dependent methods when further exploring bioactive components from some microorganisms.

Some pathogenic bacteria and *Plasmodium* parasites, especially drug-resistant strains, pose a serious threat to human public health now [5]. The global problem of antibiotic resistance is fast becoming one of the major scientific issues of modern times [6]. Moreover, it does not seem to be a global problem, but in fact, malaria is impacting a considerable proportion of the population. Even now, *Plasmodium* parasites are present in eighty-five countries spread across five WHO regions [7]. Worse yet, *P. falciparum* has developed resistance to a variety of antimalarial medications, including chloroquine and artemisinin derivatives. Therefore, the development and exploitation of effective antibacterial and antiplasmodial resources and lead compounds remains an urgent need for the anti-infective drug industry.

Fungi, particularly those inhabiting a special eco-environment, have been proven as promising and prolific sources of bioactive secondary metabolites with potent anti-infectious applications [8]. In recent years, more and more research articles have reported works on the discovery of antibacterial or antimalarial natural products from marine-derived fungi [9,10,11]. However, there are few reports on the study of anti-infective medicinal sources of microorganisms in other special eco-environments. Hence, broadening the research visibility into antibacterial and antimalarial microbes in other special habitats, such as polar, alpine, hyperhaline, plant rhizosphere, animal internal habitats and so on, will be meaningful for the more comprehensive utilization of anti-infective fungal resources in special eco-environments.

The legume *Astragalus* is an important medicinal plant with a noticeable rhizosphere effect in traditional Chinese medicine and that thrives in high latitudes [12]. Although located at a low latitude, some northwestern regions of Yunnan Province in China have similar climatic characteristics as high latitudes because of the effect of high altitude (>3000 m). During our latest field investigation of the characteristic plant resource distribution of Yunnan, a total of around 30 species of *Astragalus* plants were found to grow endemically in northwest Yunnan, with altitudes ranging from 3040 to 4479 m. In addition to our previous study on the diversity of the rhizosphere microbial communities associated with three species of *Astragalus* using metagenomics [13], and the new species of rhizofungus *Varicosporellopsis shangrilaensis* isolated from the plant *A. polycladus* [14], there is still a large amount unknown about these *Astragalus*-derived microorganisms. But what should not be ignored is that the microbes in the rhizosphere of these characteristic *Astragalus* species might be abundant and produce secondary metabolites with excellent bioactivity due to the dual action of the rhizosphere and high–cold conditions. Herein, it is reasonably suspected that the rhizofungi derived from *Astragalus* plants in the alpine environment of northwestern Yunnan should have promising potential for the development and exploitation of antibacterial and antimalarial medical resources.

In this study, we isolated and identified rhizofungi from three species of *Astragalus* plants, including *Astragalus acaulis* (Aa), *A. forrestii* (Af) and *A. ernestii* (Ae) in the special high–cold environment of Yunnan at different altitudes ranging from 3225 to 4353 m; analyzed the diversity and structure of the culturable fungal communities compared with the previously reported metagenomic results [13]; conducted an anti-infective (antibacterial activity against seven Gram-positive pathogenic bacteria and antimalarial effect on two strains of *Plasmodium falciparum*) screening of the ethyl acetate (EA) crude extracts of the cultivable fungi; and performed a further isolation, structure elucidation and anti-infective evaluation of the obtained pure compounds from *Aspergillus calidoustus* AA12. Overall, through the research above, the species diversity and anti-infective medicinal value of cultivable rhizofungi isolated from Aa, Af and Ae in the special high–cold environment of northwest Yunnan can be recognized well. The information on the cultivated fungal diversity and distribution in the three species of *Astragalus* is very scarce and important, which provides a reference for the overall awareness of the correlation between the host plants, ecological environment and community composition of fungal microbes in different *Astragalus* plants growing in northwestern Yunnan. The results of anti-infectious evaluation and chemical investigation of these rhizofungi, meanwhile, suggested that ten strains in five species, such as *Aspergillus calidoustus*, *A. tabacinus*, *Penicillium brevicompactum*, *Lecanicillium aphanocladii* and *Umbelopsis nana*, should be considered as potent resources of anti-infective natural products, which is being instructive on exploring these distinctive fungi as bioactive microbial resource in special eco-environment.

## 2. Materials and Methods

### 2.1. Rhizosphere Soil Samples Collection

Soil samples were collected from the rhizosphere sediments (at depths of 4.0–5.0 cm) of three species of *Astragalus* plants in Dêqên county, Shangri-La city, Dêqên Tibetan Autonomous Prefecture, Yunnan Province, China. The primary rhizosphere samples for each *Astragalus* species are a homogeneous mixture of rhizosphere sediments from five plants, prepared in triplicate. The plants of *Astragalus acaulis* (Aa) thriving in alpine grassland and *A. ernestii* (Ae) surviving in the steep slopes of flowstone beaches were distributed in Baima Snow Mountain at, respectively, 4353 and 4048 m of altitude (99°2′ E, 28°21′ N). The plant *A. forrestii* (Af) was surrounded by more pine trees and deciduous foliage inhabits in Xiaozhongdian town in Shangri-La at an elevation of 3225 m (99°49′ E, 27°26′ N). The detailed sampling site and plant appearance, growing environment and soil physicochemical properties of these three *Astragalus* are shown in Appendix A and Appendix A, respectively [13]. These three healthy plants and the soil closely attached to the roots (within 2.5 mm) [15] were randomly collected in each plot. These samples packed in aseptic bags were immediately transported at low temperature to the laboratory and stored at 4 °C until processing.

### 2.2. Isolation and Identification of Fungi

Fungal strains were isolated from the three *Astragalus* rhizosphere soils using a standard dilution-plating technique. About 1.5 g soil sample was weighed and suspended in 15 mL of sterile distilled water and then diluted to the final concentration of 10^−3^ g/mL using the gradient dilution method. About 200 μL of soil suspension was evenly spread on the surface of potato dextrose agar (PDA, Vokai Biochimical Co., Ltd., Beijing, China) plates with 100 mg/L of streptomycin sulfate and penicillin (Macklin Biochimical Co., Ltd., Shanghai, China) [14]. The plates were incubated at 28 °C for 3–5 days until several individual colonies formed and were then observed with a stereomicroscope (Paidiwei Instrument Co., Ltd., Beijing, China). The fungal mycelia were transferred to fresh PDA plates using a sterile needle. This step was repeated until the pure culture was obtained.

Cultivable fungal genomic DNA was extracted using a Rapid Genomic DNA Isolation Kit (Sangon Biotech Co., Ltd., Shanghai, China) according to the manufacturer’s instructions. Polymerase Chain Reaction (PCR) amplification was performed using the primers ITS1(5′-TCCGTAGGTGAACCTGCGG3′) and ITS4(5′-TCCTCCGCTTATTGATATGC-3′) for the extracted fungal genome. The PCR amplification was performed in a 25 μL reaction system containing 2 μL of DNA template, 1 μL of each primer, 8.5 μL of ddH_2_O and 12.5 μL of 2 × Taq PCR MasterMix (Sangon, Biotech Company, Limited, Shanghai, China). PCR amplifications were performed using the following program: after pre-denaturation at 94 °C for 10 min, amplification was performed with 30 cycles, followed by denaturation at 94 °C for 30 s, annealing at 55 °C for 30 s, extension at 72 °C for 90 s and extension at 72 °C for 5 min. The PCR products were verified by agarose gel electrophoresis. All sequencing results were analyzed for shearing using SeqMan 8.0 software, and then the isolates were identified by the blast program in the NCBI database. The identified strains were preserved in the Yunnan Key Laboratory of Screening and Research on Anti-pathogenic Plant Resources from Western Yunnan, Dali, Yunnan, China.

### 2.3. Fermentation and Extraction

All fungal isolates were cultivated in solid rice medium (100 mL double-distilled water, 100 g rice, 0.6 g peptone) in a 1 L Erlenmeyer flask at 27 °C for 4 weeks. Ethyl acetate (EA, Huiyun Chemical Store, Dali, Yunnan, China) was used to extract repeatedly (3 × 300 mL) the fermented substrate. Then, the extractive EA was concentrated by vacuum evaporation to afford a residue. Finally, the EA crude extract of the fungal fermentation was obtained and dissolved in indimethyl sulfoxide (DMSO, Aladdin Biochimical Co., Ltd., Shanghai, China) to afford 100 mg/mL stock solution for testing.

### 2.4. Antibacterial Bioassay

A total of seven strains of Gram-positive pathogenic bacteria, including methicillin-resistant *Staphylococcus aureus* (MRSA) ATCC 43300, *S. aureus* ATCC 25923, *S. epidermidis* ATCC 12228, *Streptococcus agalactiae* ATCC 13813, vancomycin-resistant *Enterococcus faecalis* ATCC 51299, *E. faecalis* ATCC 29212 and *E. faecium* ATCC 35667, were used for antibacterial bioassays. The method of antibacterial evaluation (represented by MIC value) was in accordance with that described by Song et al. [16]. Specifically, these pathogens were incubated in Lysogeny broth (LB) at 28 °C and collected at the OD_595_ value in a range of 0.1 to 0.2. Then, the culture collection was diluted by LB medium to obtain a final bacterial suspension of 5 × 10^−5^ CFU/mL, 200 µL of which was transferred into a 96-well microplate (Corning, Corning, NY, USA). The desired stock solution of samples was added to each well containing 100 µL microbial suspension and further diluted using a serial two-fold dilution to obtain the tested solutions with five concentrations (primary screening concentrations of 100, 50, 25, 12.5 and 6.25 μg/mL, and secondary screening concentrations of 25, 12.5, 6.25, 3.125 and 1.5625 μg/mL). There were five replicate wells per concentration. The wells containing DMSO–suspension (*v*/*v* 1:100) and 2 μg/mL of vancomycin served as the negative and positive controls, respectively. These tested microplates were then incubated at 37 °C and the OD_595_ value was measured after 24 h for the determination of the MIC value. All the experiments were repeated three times.

### 2.5. Antimalarial Bioassay (In Vitro)

Two strains, *Plasmodium falciparum* of 3D7 and chloroquine-resistant Dd2, were used as model protozoa, which were provided by Shanghai Institute of Immunity and Infection, Chinese Academy of Sciences. Following the modified protocol stated by Gudla et al. [17] with few differences, *Plasmodium* strains were maintained in RPMI-1640 medium at 2–5% hematocrit supplemented with 0.2% *w*/*v* glucose, 0.5% *w*/*v* AlbuMaxII, 0.22% *w*/*v* NaHCO_3_, 50 µg/mL gentamycin and 50 µg/mL hypoxanthine and incubated at 37 °C for 72 h under a micro-aerophilic atmosphere (4% CO_2_, 3% O_2_ and 93% N_2_). The 5% *w*/*v* D-sorbitol method was used for parasite synchronization. The stock solution of the tested samples was serially diluted two-fold with culture medium. A total of 10 μL of the diluted substance was added to 190 μL parasite inoculum (1% parasitaemia, 2% haematocrit) in a 96-well tissue culture plate. After 72 h of incubation at 37 °C, each well was further processed by 100 µL of RBCs lytic buffer (5 mM EDTA, 20 mM Tris pH 7.5, 0.12% *v*/*v* Triton X-100 and 0.012% *w*/*v* Saponin) containing the SYBR green 1× final concentration. Then, plates were incubated at room temperature for 2 h in the dark and measurement conducted under a fluorescence reader at an excitation of 485 nm and an emission of 535 nm. The groups of DMSO–suspension (*v*/*v* 1:100) and chloroquine (0.16 μg/mL)-treated samples served as the negative and positive controls, respectively. The inhibition rate for the antimalarial activity of the tested samples was determined based on the DNA content of the parasites. The IC_50_ value for *Plasmodium* was calculated by GraphPad Prism 8.0 software. Differences in all tests between treatments and controls were determined by one-way ANOVA followed by Tukey’s test. Results from the one-way ANOVA were reported as means and standard deviations (SD) and considered to be significant when *p* < 0.05. Each bioassay experiment had five biological replicates and was repeated three times using three different batches of *Plasmodium.*

### 2.6. Isolation and Structural Elucidation of Secondary Metabolites from Aspergillus calidoustus AA12

The fungus *A. calidoustus* AA12 was finally chosen as the target strain because of its excellent antibacterial and antimalarial activities. A total of fifty Erlenmeyer flasks of the fungi AA12 were cultivated. The conditions of fermentation and extraction were the same as previously mentioned in Section 2.3. About 37.2 g of EA crude extract was obtained and then subjected to the general experimental procedures of compounds’ isolation. In this study, Silica gel GF254 plates (Qingdao Marine Chemical Ltd., Qingdao, China) were used for thin-layer chromatography (TLC) analysis. Silica gel (Qingdao Marine Chemical Ltd., Qingdao, China) and Sephadex LH-20 (Amersham Biosciences, Uppsala, Sweden) were applied for column chromatography (CC). High-performance liquid chromatography (HPLC) analysis and separation was performed on Agilent 1260 system coupled with Agilent G1315D photodiode array detector using analytical (Agilent ZORBAX, 5 μm, 9.4 × 250 mm) and semipreparative (Agilent Eclipse XDB, 5 μm, 4.6 × 250 mm) C_18_ columns, respectively. The EA crude extract of fermented substances of strain AA12 was primarily subjected to vacuum liquid chromatography (VLC) on silica gel CC using step gradient elution with EA–petroleum ether (PE) (0–100%) and then with methanol (MeOH)–EA (0–100%) to yield eight fractions (Fr. 1–Fr. 8). Fr. 2 was isolated by silica gel CC (PE–EA, *v*/*v* 10:1–7:3), then applied to silica gel CC eluted by chloroform (CHCl_3_)–MeOH (*v*/*v* 80:1−50:1) and repeated Sephadex LH-20 CC elution with MeOH and further purified on HPLC (85% MeOH–H_2_O) to obtain the compounds **1** (267 mg) and **2** (238 mg). Fr. 3 was primarily fractionated on silica gel CC (PE–EA, *v*/*v* 9:1–1:1) to afford four subfractions (Fr. 3-2–Fr. 3-4). Fr. 3-2 was subjected to silica gel CC (PE–acetone, *v*/*v* 4:1–1:1) and repeated Sephadex LH-20 CC (CHCl_3_-MeOH, *v*/*v* 1:1) and further purified on HPLC (87% MeOH–H_2_O) to afford the compounds **3** (18 mg) and **4** (31 mg). Fr. 3-3 was isolated by silica gel CC (CHCl_3_–MeOH, *v*/*v* 50:1–40:1) to obtain the compound **5** (13 mg) and then purified on HPLC using step gradient elution with MeOH–H_2_O (55–70%) to give the compounds **6** (15 mg) and **7** (4.9 mg). The structure of compounds were elucidated by NMR spectra, which were recorded on a Bruker Avance III-400 instrument (Bruker, Faellanden, Switzerland) with TMS as an internal standard.

### 2.7. Scanning Electron Microscopic (SEM) Observation

To visually disclose the inhibition of 6-epi-ophiobolin G on *Streptococcus agalactiae* biofilm, the alterations in superficial structure and morphology of biofilm induced by 6-epi-ophiobolin G was observed by scanning electron microscope (SEM). According to the reported method with few modifications [16,18], a total of 400 μL logarithmic phase *S. agalactiae* ATCC 13813 cell suspension and 600 μL different concentrations of tested samples were added into each well of 24-well polystyrene plate (Nunc, Naperville, CA, USA). After incubation at 37 °C for 10 h, the biofilms of each well was rinsed with PBS and then fixed with glutaraldehyde for 10 h at 4 °C, followed by dehydration with an increasing gradient of 30%, 50%, 70%, 90% and 100% ethanol at 4 °C for 10 min of each concentration. Finally, each processed sample was transferred to the circular slide, air dried and sputtered by coating with gold for 50 s. SEM imaging was performed on a JSM-7500F scanning electron microscope (JEOL, Tokyo, Japan).

### 2.8. Nucleotide Sequence Accession Number

The ITS sequences of all 93 fungal isolates obtained in this experiment were registered in GenBank. The accession numbers are PP380353–PP380390.

## 3. Results and Discussion

### 3.1. Diversity of Cultivable Fungi in the Rhizosphere Communities of Three Species of Astragalus Plants

The diversity of culturable fungi in the rhizosphere communities of *Astragalus acaulis* (Aa), *A. forrestii* (Af) and *A. ernestii* (Ae) was determined at the genus and species level. A total of 93 fungal strains were isolated from the rhizosphere soil of Aa (34 isolates), Af (40 isolates) and Ae (19 isolates) based on the size, color and other morphological observations of fungi and identified according to the BLASTn results of ITS sequences, including 15 species of 12 genera for Aa, 19 species of 6 genera for Af and 12 species of 9 genera for Ae (Table 1). In general, these 93 isolates belong to 38 different species in 18 genera sharing a high similarity (98–100%) of the ITS-rDNA sequences to their closest NCBI relatives (Table 1), which showed the great diversity of fungal communities in the rhizosphere of the plants Aa, Af and Ae. Furthermore, the most common trend of a decrease in biodiversity with increasing altitude observed in community ecology does not seem to occur in the rhizosphere fungal communities of Aa, Af and Ae with the elevations of 4353, 3225 and 4048 m, respectively. This phenomena was also observed in our previous study of the microbial diversity of *Astragalus* plants based on metagenomics [13]. As reported by Ding et al. [13], soil physicochemical properties, plant species, the period of plant growth and some other factors have a great aggregate effect on the community structure of rhizosphere microbes.

There are four genera (Figure 1A, including *Penicillium*, *Aspergillus*, *Mortierella* and *Trichoderma*) and two species (Figure 1B, including *A. fumigatus* and *M. alpine*) common to the 18 genera and 38 species identified. Among them, the top two dominant genera are *Penicillium* and *Aspergillus*, which is aligned with the general idea that both genera tend to be common in soil worldwide [19,20]. Some members of these two genera are known as biocontrol fungal microbes and for their ability to produce large amounts of functional secondary metabolites with potent bioactive applications [21,22], hence justifying their highest abundance in the rhizosphere fungal communities of these three *Astragalus* plants growing in harsh environments. The genus *Mortierella* was the third most abundant genera, of which the species *M. alpine*, as the most dominant population, was commonly distributed in each *Mortierella* rhizosphere community of the three types of *Astragalus*. This finding was consistent with the previous research which reported the dominant distribution of the *Mortierella* in the soil samples of Tibet [23] and the Greater Khingan Mountains [24], China. Therefore, it is reasonable to presume that the species of *M. alpine* may prefer to inhabit in the special high–cold environment, and herein this species can be considered as the dominant cultivable rhizofungi with distinctly eco-environmental characteristics for these three *Astragalus*. Meanwhile, a total of 7, 2 and 4 genera (Figure 1A and Appendix A) and 10, 15, 7 species (Figure 1B and Appendix A) were uniquely distributed in the rhizosphere fungi of Aa, Af and Ae. At the genus level, the rhizosphere fungi derived from Aa and Ae has better species uniqueness than those of Af. For instance, the *Pseudogymnoascus roseus* originally discovered in Antarctica [25] and the *Samsoniella hepiali* mainly from *Ophiocordyceps sinensis* distributing in the the plateau meadow of northwest Tibet, Qinghai, Sichuan, and Yunnan at 3000–8000 m of altitude [26] was only found in the rhizosphere of Aa group. This may be explained by the fact that the host plant Aa survive in the high altitude and low heat environment at an elevation of 4353 all year round. In addition, *Aporospora terricola*, as the unique fungal species, was present in the Ae group and had ever been repeatedly isolated from the the extremely arid desert nature reserve in Anxi, Gansu province of China [27]. As shown in Appendix A [13], compared with the fertile vegetation-covered habitats of the plants Aa and Af, the plant Ae thrived in the steep slopes of flowstone beaches of 4048 m that is a typical arid and cold mountain ecosystem, which has some similarities to that of the desert ecosystem. But in the end, there are something cannot be neglected; that is, many dominant and unique fungi referred to in the previously metagenomic study on the microbial diversity of rhizosphere soil of these three *Astragalus* plants [13] were not obtained and identified in this investigation, indicating that the selection of culture medium and conditions for these fungal microbes should be more diverse and specific.

### 3.2. Anti-Infectious Potential of Isolated Fungi

All 93 fungal isolates were fermented on a small scale in solid rice medium and the EA crude extracts of their fermented substances were evaluated for antibacterial and antimalarial activities. In total, ten extracts showed weak to strong antibacterial activity against seven Gram-positive pathogenic bacteria with MIC values ranging from 100 to 6.25 µg/mL (Table 2), and among which three extracts could inhibit the growth of 80–99% *Plasmodium falciparum* 3D7 at the concentration of 50 µg/mL (Table 3). The micro-morphological characteristics of the colonies of the ten fungi have been presented in Appendix A. Of these, ten bioactive fungal strains, about 70% species belonged to the genera *Aspergillus* and *Penicillium* that is the dominant flora in the cultivated fungal communities of Aa, Af and Ae rhizosphere. Although many species of these two genera exist extensively in the nature and have been well explored as potential medicinal resources for producing novel and bioactive natural products [21,22], there are few reports on the detailed research of natural products of *A. calidoustus*, *A. tabacinus*, *P. glabrum* and *P. vasconiae*. In addition, to our best knowledge, the study on the secondary metabolites of the other three anti-infective species of *Cordyceps farinose*, *Lecanicillium aphanocladii* and *Umbelopsis nana* has thus far not been reported. These results of the primary anti-infective screening of fungi and the related literature investigation suggested that some fungal strains affiliating to the genera of *Aspergillus*, *Penicillium*, *Cordyceps*, *Lecanicillium* and *Umbelopsis* derived from these three *Astragalus* rhizosphere might be a potent source of anti-infective natural products.

Most notably, the EA crude extract of *Aspergillus calidoustus* AA12 fermented substance isolated from the plant Aa rhizosphere showed both excellent antibacterial and antimalarial activities. Specifically, the extract of strain AA12 revealed the strongest antibacterial against methicillin-resistant *Staphylococcus aureus* (MRSA) ATCC 43300, *S. aureus* ATCC 25923 and *Enterococcus faecalis* ATCC 29212 with the MIC values of 6.25 µg/mL and showed significant inhibition for *S. epidermidis* ATCC 12228, vancomycin-resistant *Enterococcus faecalis* (VREF) ATCC 51299 and *E. faecium* ATCC 35667 with the MIC values of 12.5 µg/mL and *Streptococcus agalactiae* ATCC 13813 with the MIC value of 25 µg/mL. Also, this extract showed the best antimalarial activity against *Plasmodium falciparum* 3D7 with the inhibitory rate reaching up to 99% under the concentration of 50 μg/mL at 72 h. The species of *A. calidoustus* was firstly proposed as a new species in *Aspergillus* section Usti by Varga et al. [28]. Valiante et al. [29] suggested that austinoid gene clusters involved in the biosynthesis of fungus *A. calidoustus* should lead to the production of various austinoid-type meroterpenoids. Mo et al. [30] and Zhang et al. [31] found that the isolates of *A. calidoustus* from the wetland soil of Dianchi Lake in Kunming City and East Lake in Wuhan City could be able to produce meroterpenoids and drimane sesquiterpenoids, respectively, which are a prominent group of natural products and have good biological and chemical diversity [32]. In this study, the strain *A. calidoustus* AA12 was eventually selected as the target fungus for further exploitation of antibacterial and antimalarial natural products according to its prominent anti-infectious activity and chemical abundance (the HPLC spectra of the EA crude extracts of strain AA12 and other nine anti-infective fungal strains as shown in Appendix A).

### 3.3. Structural Elucidation and Anti-Infective Activity of the Isolated Natural Products

In total, seven compounds were isolated from *Aspergillus calidoustus* AA12, including 6-epi-ophiobolin G (**1**) [33], penicisochroman A (**2**) [34], pergillin (**3**) [35], 7-methyl-2-(1-methylethylethlidene)-furo [3,2-H]isoquinoline-3-one (**4**) [36], trypacidin (**5**) [37], 1,2-seco-trypacidin (**6**) [38] and questin (**7**) [39] (Figure 2), respectively, by comparing their spectroscopic data with the available literature (the ^1^H and ^13^C NMR spectra and data of these seven compounds are shown in Appendix A and Appendix A, respectively). As shown in Figure 2, the chemical structure types of these compounds comprise sesterterpene (**1**), sesquiterpenoids (**2**–**4**) and polyketides (**5**–**7**). In this study, all of these seven compounds were firstly discovered from the species of *A. calidoustus*, which indicated that the fungus *A. calidoustus* AA12 might have a huge biosynthetic capacity to produce structural diverse natural products besides the previously mentioned austinoid-type meroterpenoids [29,30] and drimane sesquiterpenoids [31]. Moreover, herein, the terpenoids **1**–**4** obtained from *A. calidoustus* of the plant *Astragalus acaulis* rhizosphere surviving in snow mountains at high elevations up to 4353 m were previously reported as the main secondary metabolites of marine or plant rhizosphere-derived fungi [33,34,36], which suggested that these sesterterpenes and sesquiterpenoids would be more likely to be biosynthesized by those fungal microorganisms in a special eco-environment.

The antibacterial activity of the compounds **1**–**7** were all evaluated. As shown in Table 4, all compounds displayed weak to strong antibacterial activity against seven strains of Gram-positive pathogens at MIC values in the range of 6.25 to 100 µg/mL except the 1,2-seco-trypacidin (**6**). Notably, the sesterterpene 6-epi-ophiobolin G (**1**) exhibited a broad-spectrum antibacterial activity against all pathogens tested with MIC values ranging from 6.25 to 50 µg/mL and showed the most obvious inhibitory effect on the growth of *Streptococcus agalactiae* ATCC 13813 at the concentration of 6.25 µg/mL. As the SEM imaging observed (Figure 3), after treatment with 1/2×MIC (3.125 µg/mL) of the compound **1**, a collapse of the *S. agalactiae* biofilm architecture occurred in which small cell clusters and short chains were loosely attached to the surface of glass slides with little debris. Further drastic damage on the *S. agalactiae* biofilm was obviously observed as the phenomenon that the total *S. agalactiae* biofilm biomass was significantly reduced when treated with the compound **1** at the MIC value of 6.25 μg/mL. Bacterial biofilms are well-known as one of the contributing factors favoring the growth of bacteria resistant to antibiotics [9,16,18]. The pathogenic *S. agalactiae* poses a serious threat to high-risk groups such as pregnant women, newborns, and the elderly, causing severe clinical illnesses and increasing morbidity and mortality rates; and some pathogenic *S. agalactiae* have become increasingly resistant to second-line antibiotics like erythromycin and clindamycin [40]. Nowadays, the clinical treatment of *S. agalactiae* is more challenging with the emergence of penicillin-resistant strains [41]. Thus the remarkable disintegration efficacy of 6-epi-ophiobolin G (**1**) against *S. agalactiae* biofilm indicates that this compound might be potentially valuable in controlling the drug-resistant properties of *S. agalactiae*.

Although the antimalarial activity of the compounds **1**–**7** against the strains 3D7 and chloroquine-resistant Dd2 of *Plasmodium falciparum* has been detected, only the sesterterpene 6-epi-ophiobolin G (**1**) showed good antimalarial activity against the strains 3D7 and Dd2 with IC_50_ values of 3.319 and 4.340 µmol/L at 72 h, respectively (Figure 4). In addition, as shown in Figure 5, the compound **1** could effectively inhibit the period of *Plasmodium* trophozoites in order to achieve an excellent antiplasmodial effect.

In this study, 6-epi-ophiobolin G (**1**), as the most potent anti-infective sesterterpene compound isolated from *Aspergillus calidoustus* AA12, belongs to the ophiobolins, which are a prominent group of natural products produced by fungi derived from pathogenic plants [42,43], mangrove [44] and marine sediments [45]. This compound was successively isolated from the fungus *Emericella variecolor* GF10 derived from marine sediment collected at a 70 m depth in the Gokasyo Gulf, Mie Prefecture, Japan [45], and the deep-sea derived fungus *Aspergillus* sp. WHU0154 at a 3197 m depth, in the South China Sea [33], and only evaluated for its anti-inflammatory effect [33]. Therefore, it is the first report that the ophiobolin type was isolated from a fungal microbe in a high–cold terrestrial environment of a snow mountain and proved to have potent anti-infective activity against the biofilm of *Streptococcus agalactiae* ATCC 13813, *Plasmodium falciparum* 3D7 and chloroquine-resistant *P. falciparum* Dd2. These results will certainly provoke some inspiring action in the development and exploitation of ophiobolin-type sesterterpenoids with antibacterial and antimalarial activities against these drug-resistant strains.

## 4. Conclusions

This is the first report on the species diversity in cultivable rhizofungi derived from *Astragalus* plants in special high –cold environments and their antibacterial and antimalarial potential. The fungal communities in the rhizosphere of *A. acaulis*, *A. forrestii* and *A. ernestii* had a rich diversity and uniqueness, the microbial distribution and structure of which was influenced by a complex mixture of factors such as the altitude, soil physicochemical property, plant species and so on. Some fungal strains of *Aspergillus* and *Penicillium* displayed good anti-infective activity and chemical diversity, which might not only play an important role in helping their host *Astragalus* to grow healthily and enhance its resilience and adaptability in extremely harsh environments of high altitudes, but also provide a rich resource for our further research on bioactive natural products from *Astragalus* rhizofungi. Interestingly enough, the compounds of sesterterpine (**1**), sesquiterpenes (**2**–**4**) and polyketides (**5**–**7**) were firstly discovered in the species *Aspergillus calidoustus,* indicating the potent production of diverse chemical structural secondary metabolites of *A. calidoustus* AA12. Meanwhile, herein, the broad-spectrum antibacterial activity and excellent antimalarial activity of 6-epi-ophiobolin G (**1**) was first proposed, which may be valuable, to some extent, for further exploring and utilizing anti-infectious natural products from characteristic microbes in special high–cold environments.

## Figures and Tables

**Figure 1 jof-10-00736-f001:**
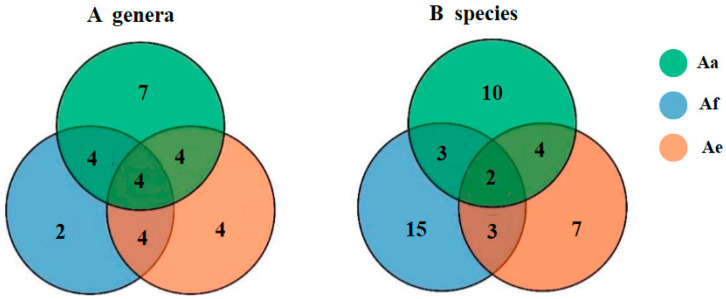
Venn diagram showing the number of fungi at genus (**A**) and species (**B**) level in the rhizosphere soil samples of *Astragalus acaulis* (Aa), *A. forrestii* (Af) and *A. ernestii* (Ae). Each circle, with a different color in the diagram, represents the number of genera and species specific to the corresponding subgroup. Middle core numbers represent the number of genera and species commonly to all groups.

**Figure 2 jof-10-00736-f002:**
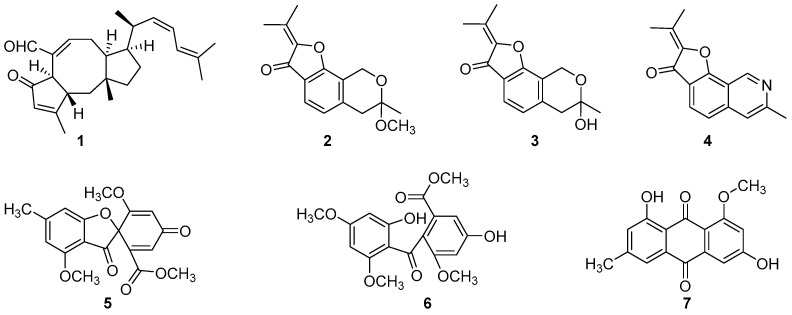
Chemical structures of the compounds **1**–**7**.

**Figure 3 jof-10-00736-f003:**
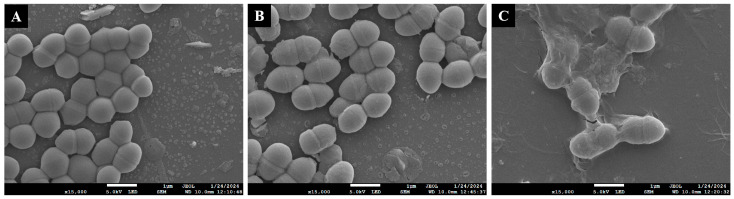
Scanning electron micrographs (SEM) ×15,000 of 24 h preformed *Streptococcus agalactiae* ATCC 13813 biofilms treated with different concentrations of 6-epi-ophiobolin G (**1**). (**A**–**C**) Different treatment groups ((**A**), DMSO control; (**B**), treated with 1/2 × MIC = 3.125 μg/mL of the compound **1**; (**C**), treated with MIC = 6.25 μg/mL of the compound **1**).

**Figure 4 jof-10-00736-f004:**
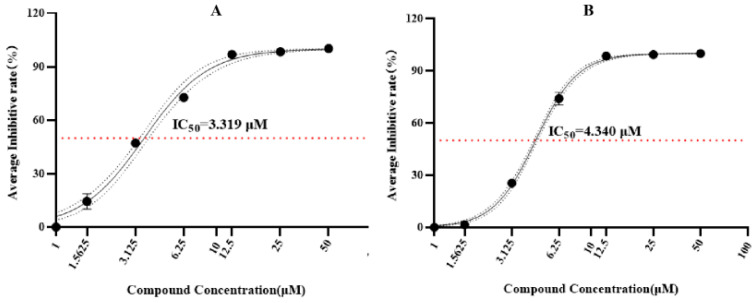
Determination of the antimalarial activity of 6-epi-ophiobolin G (**1**) at 72 h against *Plasmodium falciparum* 3D7 (**A**) and chloroquine-resistant *P. falciparum* Dd2 (**B**).

**Figure 5 jof-10-00736-f005:**
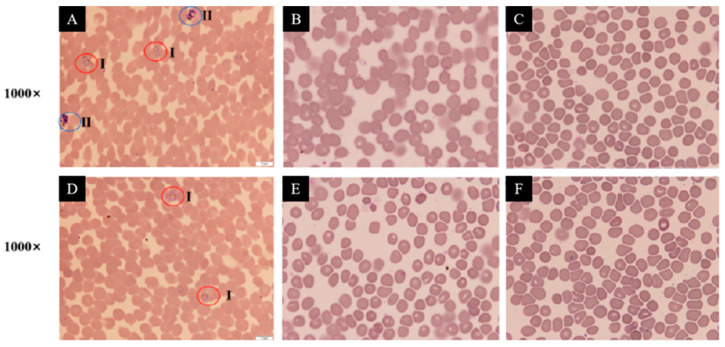
Blood smear of *Plasmodium falciparum* 3D7 and chloroquine-resistant *P. falciparum* Dd2 treated by different concentrations of 6-epi-ophiobolin G (**1**). The pictures of A1–A3 belong to the *P. falciparum* 3D7 tested group including a DMSO control (**A**) and treated with 50 µmol/L (**B**) and 25 µmol/L (**C**) of the compound **1**. The pictures of (**D**–**F**) belong to the chloroquine-resistant *P. falciparum* Dd2 tested group including a DMSO control (**D**) and treated with 50 µmol/L (**E**) and 25 µmol/L (**F**) of the compound **1**. I and II represent the periods of the trophozoite and schizont of *Plasmodium* parasites, respectively.

**Table 1 jof-10-00736-t001:** Diversity of the cultivable rhizosphere fungi from *Astragalus acaulis* (Aa), *A. forrestii* (Af) and *A. ernestii* (Ae). A total of 93 fungal strains belonging to 38 species in 18 genera were identified by comparison their internal transcribed spacer (ITS) sequences of the isolates with their reference strains in the GeneBank of NCBI. All these isolates have high identity percentage values with 98–100%.

Species	Representative Isolates (Accession Number in GenBank)	Similarity	Number of Fungal Isolates		
			Aa	Af	Ae
*Penicillium wellingtonense*	AA3 (JN617713)	100.00%	2		
*P. polonicum*	AA17 (OK510248)	99.19%	1		2
*P. glabrum*	AE3-1 (OP179016)	99.49%		1	2
*P. brevicompactum*	AE7 (MT558924)	100.00%			1
*P. vasconiae*	AF5 (OQ870819)	98.86%		8	
*P. suaveolens*	AF6 (MH864249)	99.48%		1	
*P. thomii*	AF19 (OR574374)	99.58%		1	
*Aspergillus calidoustus*	AA12 (MZ267040)	100.00%	3		
*A. fumigatus*	AA15 (MG991598)	100.00%	2	9	2
*A. niger*	AE10 (OM841707)	99.74%			1
*A. versicolor*	AF2-1 (MT582751)	100.00%		1	
*A. tabacinus*	AF16 (OK011797)	98.68%		2	
*Mortierella alpina*	AA10 (MT366045)	100.00%	8	2	3
*M.clonocystis*	AA2 (LC515184)	100.00%	3		
*M. minutissima*	AF9 (MT072066)	98.93%		1	
*M. verticillata*	AF20 (MT028128)	99.50%		1	
*Truncatella angustata*	AA1-1 (MT514384)	100.00%	3		
*Cordyceps farinosa*	AA8 (MF494607)	99.82%	3		
*Exophiala tremulae*	AA21 (NR159874)	98.84%	1		
*Beauveria pseudobassiana*	AA4-1 (KT368174)	100.00%	1		
*Samsoniella hepiali*	AA16 (OL684609)	98.91%	2		
*Cladosporium cladosporioides*	AA2-1 (ON970159)	100.00%	1		2
*Neonectria radicicola*	AA1 (AJ875331)	100.00%	1		
*Trichoderma atroviride*	AA14 (KX379158)	99.63%	1	3	
*T. paraviridescens*	AF8 (MN900599)	100.00%		1	
*T. koningiopsis*	AF23 (KU645324)	99.83%		2	
*T. viride*	AF29 (KU202222)	98.71%		1	
*T. longipile*	AF30 (AY737763)	100.00%		2	
*T. longibrachiatum*	AE8 (KU945846)	99.71%			1
*Pseudogymnoascus roseus*	AA4 (KJ755524)	100.00%	2		
*Leptosphaeria sclerotioides*	AE3 (OR782803)	99.76%			2
*Fusarium solani*	AE4-1 (KT192216)	98.98%			1
*Nemania diffusa*	AE4 (MK336457)	99.21%			1
*Aporospora terricola*	AE1 (KC292841)	98.47%			1
*Lecanicillium aphanocladii*	AF15 (OR752316)	99.32%		1	
*Umbelopsis vinacea*	AF14 (KT354998)	99.34%		1	
*U. ramanniana*	AF26 (GU934569)	98.21%		1	
*U. nana*	AF31 (OR724064)	100.00%		1	

**Table 2 jof-10-00736-t002:** Antibacterial screening of rhizosphere fungi associated with *Astragalus acaulis* (Aa), *A. forrestii* (Af) and *A. ernestii* (Ae) against seven strains of Gram-positive pathogenic bacteria, including methicillin-resistant *Staphylococcus aureus* (MRSA) ATCC 43300, *S. aureus* ATCC 25923, *S. epidermidis* ATCC 12228, vancomycin-resistant *Enterococcus faecalis* (VREF) ATCC 51299, *E. faecalis* ATCC 29212, *E. faecium* ATCC 35667 and *Streptococcus agalactiae* ATCC 13813 (µg/mL).

Fungal Strains	Methicillin-Resistant *Staphylococcus aureus* (MRSA) ATCC 43300	*S. aureus* ATCC 25923	*S. aureus* ATCC 12228	Vancomycin-Resistant *Enterococcus faecalis* (VREF)ATCC 51299	*E. faecalis* ATCC 29212	*E. faecium* ATCC 35667	*Streptococcus agalactiae* ATCC 13813
*Aspergillus calidoustus* AA12	6.25	6.25	12.5	12.5	6.25	12.5	25
*A. fumigatus* AA15	25	25	25	50	50	50	25
*A. tabacinus* AF16	50	25	25	-	-	-	50
*Penicillium glabrum* AE3-1	50	50	12.5	-	-	-	50
*P. polonicum* AA17	50	50	50	-	-	-	50
*P. brevicompactum* AE7	10	50	25	-	-	-	50
*P. vasconiae* AF5	-	-	-	-	-	-	50
*Cordyceps farinosa* AA8	100	100	100	-	-	-	50
*Lecanicillium* *aphanocladii* AF15	25	50	50	50	50	-	50
*Umbelopsis nana* AF31	50	50	25	50	50	-	50
Positive control: vancomycin	2	2	2	-	2	2	2

“-”: MIC > 100 µg/mL.

**Table 3 jof-10-00736-t003:** Antimalarial screening of rhizosphere fungi associated with *Astragalus acaulis* (Aa), *A. forrestii* (Af) and *A. ernestii* (Ae) against *Plasmodium falciparum* 3D7.

Fungal Strain	Test Concentration (µg/mL)	Inhibitory Rates ± SD (%)
*Aspergillus calidoustus* AA12	100	99.78 ± 0.21 **
	50	99.60 ± 0.29 **
*A. fumigatus* AA15	100	81.81 ± 1.86 *
	50	14.31 ± 0.61 *
*A. tabacinus* AF16	100	95.60 ± 0.90 **
	50	91.19 ± 1.02 *
*Penicillium glabrum* AE3-1	100	29.57 ± 0.84 *
	50	0.93 ± 0.37 *
*P. polonicum* AA17	100	−51.88 ± 1.80 *
	50	−32.53 ± 1.07 *
*P. brevicompactum* AE7	100	87.95 ± 0.73 **
	50	81.91 ± 2.30 *
*Cordyceps farinosa* AA8	100	−45.30 ± 2.00 *
	50	−15.46 ± 0.67 *
Positive control: chloroquine	0.16	99.88 ± 0.09 **

Data = mean ± SD (n = 3) * *p* < 0.05; ** *p* < 0.01 (one-way ANOVA).

**Table 4 jof-10-00736-t004:** Antibacterial activity of the compounds **1**–**7** against seven strains of Gram-positive pathogenic bacteria, including methicillin-resistant *Staphylococcus aureus* (MRSA) ATCC 43300, *S. aureus* ATCC 25923, *S. epidermidis* ATCC 12228, Vancomycin-resistant *Enterococcus faecalis* (VREF) ATCC 51299, *E. faecalis* ATCC 29212, *E. faecium* ATCC 35667 and *Streptococcus agalactiae* ATCC 13813 (µg/mL).

Compounds	Methicillin-Resistant *Staphylococcus aureus* (MRSA) ATCC 43300	*S. aureus* ATCC 25923	*S. aureus* ATCC 12228	Vancomycin-Resistant *Enterococcus faecalis* (VREF)ATCC 51299	*E. faecalis* ATCC 29212	*E. faecium* ATCC 35667	*Streptococcus agalactiae* ATCC 13813
**1**	25	12.25	12.5	50	12.5	25	6.25
**2**	-	-	-	-	-	-	100
**3**	-	-	-	-	-	-	100
**4**	-	-	-	-	-	-	50
**5**	-	-	-	-	-	-	100
**6**	-	-	-	-	-	-	-
**7**	100	-	-	-	-	-	-
Vancomycin	2	2	2	-	2	2	2

“-”: MIC > 100 µg/mL.

## Data Availability

The data of cultivable fungal strains in this study were uploaded to the NCBI database, and the accession number was PP380353-PP380390.

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
