# Peer review of "Diversity and Anti-Infectious Components of Cultivable Rhizosphere Fungi Derived from Three Species of Astragalus Plants in Northwestern Yunnan, China"

_jof, 2024, doi:10.3390/jof10110736_

Round 1

Reviewer 1 Report

Dear Authors,

I enjoyed reading your manuscript. The presented manuscript is aimed at an extremely relevant contemporary topic for researching the potential of natural sources of bioactive metabolites with anti-infective action. The emphasis is placed on micromycetes communities in the rhizosphere of three species of Astragalus (Legume family), formed in conditions of high altitude and low temperatures. The scope of the research is complex and wide-ranging: The natural areas of distribution of the Astragalus species and the specificity of the conditions in them were analyzed; Biodiversity of microfungi in their rhizospheres has been studied; 93 fungal strains were isolated and identified by molecular genetic analyses; Anti-infective biotests were conducted and on this basis, a suitable strain of the Aspergillus calidoustus species was selected, which showed the highest effect; Secondary metabolites produced by the strain were isolated and structurally elucidated, and their antibacterial (against G(+) bacteria) and antimalarial activity was determined; The specific compounds and doses that showed the highest effectiveness are indicated. The manuscript is very well written, with a rich data set skilfully interpreted and results well presented with adequate discussion. Both classical and modern molecular biological and chemical methods and techniques were used. The results obtained from the research enrich the knowledge in the field of discovery of new natural sources as bioactive and therapeutic agents.

I recommend that the manuscript  be accepted for publication without corrections in the Journal of Fungi, MDPI.

Dear Editor,

The presented manuscript is aimed at a current topic for researching the potential of natural sources of bioactive metabolites with anti-infective action.The emphasis is placed on micromycetes communities in the rhizosphere of three species of Astragalus (Legume family), formed in conditions of high altitude and low temperatures. The scope of the research is wide-ranging: The natural areas of distribution of the Astragalus species and the specificity of the conditions in them were analyzed; Biodiversity of microfungi in their rhizospheres has been studied; 93 fungal strains were isolated and identified by molecular genetics analyses; Anti-infective biotests were conducted and on this basis, a suitable strain of the Aspergillus calidoustus species was selected, which showed the highest effect; Secondary metabolites produced by the strain were isolated and structurally elucidated, and their antibacterial (against G(+) bacteria) and antimalarial activity was determined; The specific compounds and doses that showed the highest effectiveness are indicated.

In this regard, the theme is original. The results obtained from the research enrich the knowledge in the field of discovery of new natural sources as bioactive and therapeutic agents.

The manuscript is well written, with a rich data set, skillfully interpreted. Both classical and modern molecular biological and chemical methods and techniques were used. The results well presented with adequate discussion. The references are related to the topic and are mainly from the period 2015 – 2024.

I recommend that the manuscript be accepted for publication after minor correction

I have a few minor notes and comments

In Abstract

- To change the text at the beginning of the abstract because it is almost identical to that in an already published article (Ding et al., 2024). Lines 15-17

- To add "Astragalus" to the first species "acaulis", because the species are mentioned for the first time in the text, as well as to indicate for each species scientific name authorship. Line 18

- The note about adding scientific name authorship also applies to the rest of the text of the manuscript.

- When listing the seven compounds isolated, remove "three" because as described their number is violated. It is not necessary to indicate numbers of compound in the abstract. Replace "Compound 1" with the name of the compound. Line 25-28

In the Introduction

- It would be good to add the aim or a more detailed working hypothesis of the study at the end of the introduction, although the aim is stated in the abstract. It may be after the 96th line.

In Material and Methods

In 2.1.

Indicate the primary rhizosphere samples for each Astragalus species from how many plants were taken? Add agrochemical characteristic of soil samples. Lines 106-117

In 2.2.

- Specify - has an average soil sample prepared for each Astragalus species? Lines 119-120

- Use “serial dilutions-plating technique” instead of “standard dilution-plating technique”. Line 120

In 2.3.

- Specify that the rice medium is solid. Line 144

In 2.4

- “Vancomycin-resistant” should not be in “italic”. Line 153

- Enterococcus name to be abbreviated to E. faecalis ATCC 29212”. Line 154

- It is not stated which five concentrations were tested. Line 162

- The specific concentrations of Vancomycin used for the positive control are not indicated. Lines 163-164

- The literature sources for the methods used are not indicated everywhere.

In 2.5

- Redact the sentence "Five biological replicates and three biological replicates were carried out in the anti-malarial bioassay". Lines 186-187

- Is there a negative control for this test?

In Results

- The results are presented along with a discussion. Add “and Discussion”. Line 231

- The names of Astragalus species to be short - have already been mentioned. Lines 234-235

- Everywhere in the text to check the Latin names and correct them if necessary.

- Although that molecular techniques (ITS1 and ITS4 primers, such as a universal fungal barcode sequences) for the identification of the fungal isolates were applied, it is good to present brief macro- and micro- morphological characteristics of the colonies of the isolates used in the anti-infective bioassays (listed in Tables 2 and 3).

- In the header rows of Tables 2 and 4, for a clearer visual perception, it's good to add the names of the test pathogens, even though they are indicated in the titles of the tables. Lines 321; 392

- The meaning of the "-" sign is not explained. Lines 322; 393

-  It is not clear whether the antibacterial and antimalarial activity data presented in the Tables 3 and 4 are mean values ​​of replicates. Lines 322; 393

- I found no statistical treatment of the anti-infective activity data in the text or in the supplementary materials. It should be stated in both Results and Material and Methods, if available.

- Add the names of the isolated compounds 1-7 in the title of Figure 2. Line 366

- In my opinion, it is superfluous to present electron micrographs of Streptococcus agalactiae ATCC 395 at two magnifications (Figure 3). Let only those with an increase of x15000 remain. Line 394

Dear Authors,

I enjoyed reading your manuscript aimed at exploring the natural sources of bioactive metabolites with anti-infective activity. The topic is current, interesting and useful. The manuscript is well written, with a rich data set, skillfully interpreted. The results well presented with adequate discussion.

The study adds to the knowledge in the field of discovery of new natural sources as bioactive and therapeutic agents.

I recommend that the manuscript be accepted for publication after minor correction

I have a few minor notes and comments

In Abstract

- To change the text at the beginning of the abstract because it is almost identical to that in an already published article (Ding et al., 2024). Lines 15-17

- To add "Astragalus" to the first species "acaulis", because the species are mentioned for the first time in the text, as well as to indicate for each species scientific name authorship. Line 18

- The note about adding scientific name authorship also applies to the rest of the text of the manuscript.

- When listing the seven compounds isolated, remove "three" because as described their number is violated. It is not necessary to indicate numbers of compound in the abstract. Replace "Compound 1" with the name of the compound. Line 25-28

In the Introduction

- It would be good to add the aim or a more detailed working hypothesis of the study at the end of the introduction, although the aim is stated in the abstract. It may be after the 96th line.

In Material and Methods

In 2.1.

Indicate the primary rhizosphere samples for each Astragalus species from how many plants were taken? Add agrochemical characteristic of soil samples. Lines 106-117

In 2.2.

- Specify - has an average soil sample prepared for each Astragalus species? Lines 119-120

- Use “serial dilutions-plating technique” instead of “standard dilution-plating technique”. Line 120

In 2.3.

- Specify that the rice medium is solid. Line 144

In 2.4

- “Vancomycin-resistant” should not be in “italic”. Line 153

- Enterococcus name to be abbreviated to E. faecalis ATCC 29212”. Line 154

- It is not stated which five concentrations were tested. Line 162

- The specific concentrations of Vancomycin used for the positive control are not indicated. Lines 163-164

- The literature sources for the methods used are not indicated everywhere.

In 2.5

- Redact the sentence "Five biological replicates and three biological replicates were carried out in the anti-malarial bioassay". Lines 186-187

- Is there a negative control for this test?

In Results

- The results are presented along with a discussion. Add “and Discussion”. Line 231

- The names of Astragalus species to be short - have already been mentioned. Lines 234-235

- Everywhere in the text to check the Latin names and correct them if necessary.

- Although that molecular techniques (ITS1 and ITS4 primers, such as a universal fungal barcode sequences) for the identification of the fungal isolates were applied, it is good to present brief macro- and micro- morphological characteristics of the colonies of the isolates used in the anti-infective bioassays (listed in Tables 2 and 3).

- In the header rows of Tables 2 and 4, for a clearer visual perception, it's good to add the names of the test pathogens, even though they are indicated in the titles of the tables. Lines 321; 392

- The meaning of the "-" sign is not explained. Lines 322; 393

-  It is not clear whether the antibacterial and antimalarial activity data presented in the Tables 3 and 4 are mean values ​​of replicates. Lines 322; 393

- I found no statistical treatment of the anti-infective activity data in the text or in the supplementary materials. It should be stated in both Results and Material and Methods, if available.

- Add the names of the isolated compounds 1-7 in the title of Figure 2. Line 366

- In my opinion, it is superfluous to present electron micrographs of Streptococcus agalactiae ATCC 395 at two magnifications (Figure 3). Let only those with an increase of x15000 remain. Line 394

Author Response

Comments 1: In Abstract-To change the text at the beginning of the abstract because it is almost identical to that in an already published article (Ding et al., 2024). Lines 15-17

Response 1: Thank you for pointing this out. We have changed the expression, which is very different from that of Ding et al. (2024). Lines 15-18, Page 1.

Comments 2: In Abstract -To add "Astragalus" to the first species "acaulis", because the species are mentioned for the first time in the text, as well as to indicate for each species scientific name authorship. Line 18

Response 2: Thank you for pointing this out. We have add "Astragalus" to the first species "acaulis". Line 19, Page 1.

Comments 3: In Abstract -The note about adding scientific name authorship also applies to the rest of the text of the manuscript.

Response 3: Thank you for pointing this out. We have checked and made the similar corrections in the rest of the text of the manuscript. Line 88, Page 2.

Comments 4: In Abstract -When listing the seven compounds isolated, remove "three" because as described their number is violated. It is not necessary to indicate numbers of compound in the abstract. Replace "Compound 1" with the name of the compound. Line 25-28.

Response 4: Thank you for pointing this out. We have removed the number of compound and replaced “compound 1” with the name “6-epi-ophiobolin G” of “compound 1”. Lines 26-29, Page 1.

Comments 5: In Introduction -It would be good to add the aim or a more detailed working hypothesis of the study at the end of the introduction, although the aim is stated in the abstract. It may be after the 96th line.

Response 5: Thank you for pointing this out. We have added the aim of this research in the lines 96-98 at the page 2. Meanwhile, we think that the more detailed working hypothesis has been stated at the end of the introduction. Lines 98-108, Pages 2 to 3.

Comments 6: In 2.1- Indicate the primary rhizosphere samples for each Astragalus species from how many plants were taken? Add agrochemical characteristic of soil samples. Lines 106-117

Response 6: Thank you for pointing this out. We have indicated that the primary rhizosphere samples for each Astragalus species from the rhizosphere sediments of five plants (Lines 113-115, Page 3). Also, the agrochemical characteristic of soil samples has been provided in the Table S1 in the revised supporting information.

Comments 7: In 2.1- Specify - has an average soil sample prepared for each Astragalus species? Lines 119-120

Response 7: Thank you for pointing this out. We have stated that the rhizosphere samples for each Astragalus species were prepared in triplicate. Line 115, Page 3.

Comments 8: In 2.2- Use “serial dilutions-plating technique” instead of “standard dilution-plating technique”. Line 120

Response 8: Thank you for pointing this out. The “serial dilutions-plating technique” has been instead of “standard dilution-plating technique”. Line 128, Page 3.

Comments 9: In 2.3- Specify that the rice medium is solid. Line 144

Response 9: Thank you for pointing this out. We have specified “solid rice medium”. Line 153, Page 4.

Comments 10: In 2.4- “Vancomycin-resistant” should not be in “italic”. Line 153

-Enterococcus name to be abbreviated to E. faecalis ATCC 29212. Line 154

Response 10: Thank you for pointing this out. The “Vancomycin-resistant ” has been revised to the “non-italic” (Line 162, Page 4), and the Enterococcus name has been abbreviated to E. faecalis ATCC 29212 (Line 163, Page 4).

Comments 11: In 2.4- It is not stated which five concentrations were tested. Line 162.

Response 11: Thank you for pointing this out. We have noted the five test concentrations for the primary and seconding screening. Lines 172-174, Page 4.

Comments 12: In 2.4- The specific concentrations of Vancomycin used for the positive control are not indicated. Lines 163-164

Response 12: Thank you for pointing this out. We have noted that 2 μg/mL of vancomycin was served as the positive control. Line 175, Page 4.

Comments 13: In 2.4- The literature sources for the methods used are not indicated everywhere.

Response 13: Thank you for pointing this out. Actually, we have indicated that “The method of antibacterial evaluation (represented by MIC value) was in accordance with described by Song et al. [16].” Please see the lines 164-165 highlighted in red at the page 4.

Comments 14: In 2.5- Redact the sentence "Five biological replicates and three biological replicates were carried out in the anti-malarial bioassay". Lines 186-187

Response 14: Thank you for pointing this out. We have revised the sentence as ”Each bioassay experiment had five biological replicates and was repeated three times using three different batches of Plasmodium. ”. Lines 202-203, Page 5.

Comments 15: In 2.5- Is there a negative control for this test?

Response 15: Thank you for pointing this out. We have added the statement ” The groups of DMSO–suspension (v/v 1:100) and chloroquine (0.16 μg/mL)-treated were served as the negative and positive control, respectively.” Lines 195-196, Page 4.  

Comments 16: In Results - The results are presented along with a discussion. Add “and Discussion”. Line 231

Response 16: Thank you for pointing this out. We have added “and Discussion”. Line 251, Page 6.

Comments 17: In Results -The names of Astragalus species to be short - have already been mentioned. Lines 234-235

Response 17: Thank you for pointing this out. We have revised in the line 255 at the page 6.

Comments 18: In Results -Everywhere in the text to check the Latin names and correct them if necessary.

Response 18: Thank you for pointing this out. We have checked the Latin name of the full text and made corrections.

Comments 19: In Results - Although that molecular techniques (ITS1 and ITS4 primers, such as a universal fungal barcode sequences) for the identification of the fungal isolates were applied, it is good to present brief macro- and micro- morphological characteristics of the colonies of the isolates used in the anti-infective bioassays (listed in Tables 2 and 3).

Response 19: Thank you for pointing this out. We have noted that “the micro-morphological characteristics of the colonies of the anti-infective isolates have been presented in Figure S3” (Lines 326-327, Page 8). The Figure S3 has been provided in the revised supporting information at the page 3.

Comments 20: In Results - In the header rows of Tables 2 and 4, for a clearer visual perception, it's good to add the names of the test pathogens, even though they are indicated in the titles of the tables. Lines 321; 392

Response 20: Thank you for pointing this out. We have add the names of the test pathogens in the header rows of Tables 2 and 4. Please see the revised Table 2 (Line 344 , Page 9) and 4 (Line 419 , Pages 11-12)

Comments 21: In Results - The meaning of the "-" sign is not explained. Lines 322; 393.

Response 21: We have explained that the "-" sign means “MIC > 100 μg/mL”. Line 345, Page 9 and Line 420, Page 12.

Comments 22: In Results - It is not clear whether the antibacterial and antimalarial activity data presented in the Tables 3 and 4 are mean values of replicates. Lines 322; 393

Response 22: Thank you for pointing this out. As described in the previous sections 2.4 and 2.5, each bioassay experiment had five biological replicates and was repeated three times.

Comments 23: In Results -I found no statistical treatment of the anti-infective activity data in the text or in the supplementary materials. It should be stated in both Results and Material and Methods, if available.

Response 23: Thank you for pointing this out. The statistical treatment of the antimalarial activity data have been stated in the section 2.5 (Lines 199-201, Pages 4-5) and as the notes of Table 3.

Comments 24: In Results -Add the names of the isolated compounds 1-7 in the title of Figure 2. Line 366

Response 24: Generally speaking, according to the most research articles of natural products, the names of the isolated compounds are not suggested to present in the figure title/legend of the chemical structures. In the manuscript, the names of compounds 1-7 have been mentioned in the Section 3.3 (Lines 373-375, Page 10), we do not think it is necessary to add the names of these seven compounds in the title of Figure 2 again.

Comments 25: In Results -In my opinion, it is superfluous to present electron micrographs of Streptococcus agalactiae ATCC 395 at two magnifications (Figure 3). Let only those with an increase of x15000 remain. Line 394

Response 25: Thank you for pointing this out. Now the revised Figure 3 only presented the x 15000 electron micrographs. Line 421, Page 12.

Reviewer 2 Report

This manuscript explores the diversity and anti-infective potential of cultivable fungi isolated from the rhizosphere (root zone) of three Astragalus plants in northwestern Yunnan, China. Here's a breakdown of the key points and some suggestions for improvement:

General comments:

The abstract could be more concise and highlight the key findings.

Consider replacing jargon with simpler terms for a broader audience.

Some sentences could be rephrased for better flow (e.g., lines 38-41).

Emphasize the unique aspects of the study, such as the high-altitude environment and the first discovery of compounds from A. calidoustus.

Briefly discuss the potential applications of these findings.

Acknowledge potential limitations, such as the limitation of culture-dependent methods and the need for further studies on the identified compounds.

Consider including figure legends to explain the data presented.

Specific questions:

Line 73: Instead of "should not be ignored," consider "might be abundant and produce..."

Lines 104-117: Summarize the specific methods used for rhizosphere soil collection and fungal isolation.

Lines 143-149: Briefly explain the rationale behind using rice medium for fermentation.

Lines 183-188: Could you explain the significance of the positive control (chloroquine) in the antimalarial bioassay?

Lines 217-227: Briefly describe the SEM technique and its purpose in this study.

Author Response

Comments 1: The abstract could be more concise and highlight the key findings.

Response 1: Thank you for pointing this out. According to the combined comments of the two reviewers, we have revised the abstract.

Comments 2: Consider replacing jargon with simpler terms for a broader audience.

Response 2: Thank you for pointing this out. We have revised some expression.

Comments 3: Some sentences could be rephrased for better flow (e.g., lines 38-41).

Response 3: Thank you for pointing this out. We have rephrased the related expression. Lines 44-49, Pages 1-2. We also made other revising editing for English language and style.

Comments 4: Emphasize the unique aspects of the study, such as the high-altitude environment and the first discovery of compounds from A. calidoustus.

Response 4: Thank you for pointing this out. In the revised manuscript, “the high-altitude area” and “special high-cold environment” have been emphasized many times. And we also indicated that “all these seven compounds were firstly discovered from the species of A. calidoustus” (Line 380, Page 10).

Comments 5: Briefly discuss the potential applications of these findings.

Response 5: Thank you for pointing this out. In the introduction, we have stated briefly the potential applications of these findings (Lines 98-108, Pages 2-3), and discussed that in the lines 336-339 in the section 3.2, lines 411-414 in the section 3.3, lines 474-477 in the section of conclusions at the page 14. These findings are derived from the basic research in the laboratory. Therefore, as the revised manuscript described, the potential applications are more focused on providing some inspiration and reference to the investigation of microbes in high-altitude environment and increasing the library of fungi available for natural anti-infectious products screening.

Comments 6: Acknowledge potential limitations, such as the limitation of culture-dependent methods and the need for further studies on the identified compounds.

Response 6: Thank you for pointing this out. We discussed the limitation of culture-dependent methods at the end of the second paragraph of the section 3.1 (Lines 308-312, Page 8). In addition, the compound 6-epi-ophiobolin G (1), belonging to ophiobolin-type sesterterpenoids, is the most potent anti-infective component. Therefore, we discussed the discovery of this compound and pointed that the derivatives of 6-epi-ophiobolin G are worth for further studies (Lines 444-458, Page 13).

Comments 7: Consider including figure legends to explain the data presented.

Response 7: Thank you for pointing this out. All Figures have figure legends to explain the data presented in the revised manuscript except Figure 3.

Comments 8: Line 73- Instead of "should not be ignored," consider "might be abundant and produce..."

Response 8: Thank you for pointing this out. We have revised the expression. Line 82, Page 2.

Comments 9: Lines 104-117- Summarize the specific methods used for rhizosphere soil collection and fungal isolation.

Response 9: Thank you for pointing this out. The specific methods used for rhizosphere soil collection and fungal isolation have been described in the sections 2.1 and 2.2. We also provided the literature sources of these methods.

Comments 10: Lines 143-149-Briefly explain the rationale behind using rice medium for fermentation.

Response 10: Rice medium is a common medium for fungal fermentation, which can provide rich carbon, nitrogen and energy sources for the growth and metabolism of fungi. Therefore, we do not think it is necessary to explain why we choose rice medium for the fermentation of fungi in the manuscript.

Comments 11: Lines 183-188- Could you explain the significance of the positive control (chloroquine) in the antimalarial bioassay?

Response 11: Chloroquine is a known antimalarial drug. The chloroquine-treated group as a positive control in antimalarial experiments can verify the reliability of experimental methods and conditions by observing its inhibitory effect on malaria. In addition, the positive control results of chloroquine can also be used as a standard to compare with other experimental results to ensure consistency and comparability of experimental results. Because the chloroquine has become an conventional positive drug in the most of present antimalarial research, there is no detailed explain for the the significance of the positive control (chloroquine) in the manuscript.

Comments 12: Lines 217-227-Briefly describe the SEM technique and its purpose in this study.

Response 12: Thank you for pointing this out. We have added the statement as “To visually disclose the inhibition of 6-epi-ophiobolin G on Streptococcus agalactiae biofilm, the alterations in superficial structure and morphology of biofilm induced by 6-epi-ophiobolin G was observed by SEM. “ Lines 236-238, Page 5.

Round 2

Reviewer 2 Report

I am writing to follow up on my review of manuscript jof-3184873, entitled: Diversity and Anti-infectious Components of Cultivable Rhizosphere Fungi Derived from Three Species of Astragalus Plants in Northwestern Yunnan, China. I'm pleased to inform you that the authors have addressed all my comments thoughtfully and respectfully.

Their revisions demonstrate a clear understanding of the points I raised, and the changes they have made significantly improve the quality of the manuscript. I believe the paper is now much stronger and ready for further consideration.

I am writing to follow up on my review of manuscript jof-3184873, entitled: Diversity and Anti-infectious Components of Cultivable Rhizosphere Fungi Derived from Three Species of Astragalus Plants in Northwestern Yunnan, China. I'm pleased to inform you that the authors have addressed all my comments thoughtfully and respectfully.

Their revisions demonstrate a clear understanding of the points I raised, and the changes they have made significantly improve the quality of the manuscript. I believe the paper is now much stronger and ready for further consideration.